# Innovative Computerized Dystrophin Quantification Method Based on Spectral Confocal Microscopy

**DOI:** 10.3390/ijms24076358

**Published:** 2023-03-28

**Authors:** Anna Codina, Mònica Roldán, Daniel Natera-de Benito, Carlos Ortez, Robert Planas, Leslie Matalonga, Daniel Cuadras, Laura Carrera, Jesica Exposito, Jesus Marquez, Cecilia Jimenez-Mallebrera, Josep M. Porta, Andres Nascimento, Cristina Jou

**Affiliations:** 1Neuromuscular Unit, Hospital Sant Joan de Déu, Esplugues de Llobregat, 08950 Barcelona, Spain; 2Biobank, Hospital Sant Joan de Déu, Esplugues de Llobregat, 08950 Barcelona, Spain; 3Institut de Recerca Sant Joan de Déu, Esplugues de Llobregat, 08950 Barcelona, Spain; 4Confocal Microscopy and Cellular Imaging Unit, Genetic and Molecular Medicine Department, Pediatric Institute for Rare Diseases, Hospital Sant Joan de Déu, Esplugues de Llobregat, 08950 Barcelona, Spain; 5Institut de Robòtica i Informàtica Industrial, Technical University of Catalonia (UPC) and the Spanish Council for Scientific Research (CSIC) Llorens i Artigas 4-6, 08028 Barcelona, Spain; 6CNAG-CRG Centro Nacional de Análisis Genomico (CNAG)-Centre for Genomic Regulation (CRG), The Barcelona Institute of Science and Technology, Baldiri Reixac 4, 08028 Barcelona, Spain; 7Statistics Department, Fundació Sant Joan de Déu, Esplugues de Llobregat, 08950 Barcelona, Spain; 8Pathology Department, Hospital Sant Joan de Déu, Esplugues de Llobregat, 08950 Barcelona, Spain

**Keywords:** dystrophin, Duchenne muscular dystrophy, Becker muscular dystrophy, confocal microscopy, fluorescence quantification, spectral imaging

## Abstract

Several clinical trials are working on drug development for Duchenne and Becker muscular dystrophy (DMD and BMD) treatment, and, since the expected increase in dystrophin is relatively subtle, high-sensitivity quantification methods are necessary. There is also a need to quantify dystrophin to reach a definitive diagnosis in individuals with mild BMD, and in female carriers. We developed a method for the quantification of dystrophin in DMD and BMD patients using spectral confocal microscopy. It offers the possibility to capture the whole emission spectrum for any antibody, ensuring the selection of the emission peak and allowing the detection of fluorescent emissions of very low intensities. Fluorescence was evaluated first on manually selected regions of interest (ROIs), proving the usefulness of the methodology. Later, ROI selection was automated to make it operator-independent. The proposed methodology correctly classified patients according to their diagnosis, detected even minimal traces of dystrophin, and the results obtained automatically were statistically comparable to the manual ones. Thus, spectral imaging could be implemented to measure dystrophin expression and it could pave the way for detailed analysis of how its expression relates to the clinical course. Studies could be further expanded to better understand the expression of dystrophin-associated protein complexes (DAPCs).

## 1. Introduction

Duchenne muscular dystrophy (DMD) and Becker muscular dystrophy (BMD) are X-linked genetic neuromuscular diseases that affect 1 in 5000 new-born males. DMD is the most common type of muscular dystrophy and also one of the most severe [1]. The main clinical features of DMD are proximal muscle weakness, loss of ambulation at an early age, progressive cardiomyopathy and restrictive respiratory failure, these last two being the most common causes of early death [2]. BMD has less severe symptoms and a slower progression [3]. Both DMD and BMD are caused by hemizygous pathogenic variants in the *DMD* gene, with large deletions (60–65%) or duplications (5–15%) being the most common, but small mutations such as point mutations and deep intronic mutations also occur [2,4]. According to Monaco’s law [5], mutations disrupting the open reading frame will generally trigger an almost total absence of the dystrophin protein, resulting in a severe, DMD phenotype. In contrast, pathogenic variants that maintain the open reading frame, enabling residual protein function, result in a milder, BMD phenotype.

To date there is no curative treatment for DMD or BMD, but several therapeutic approaches are being developed, which can be divided into two broad groups: (i) those that pursue dystrophin restoration, and (ii) those that try to counteract the consequences of dystrophin deficiency. Regarding protein restoration, different therapeutic approaches such as antisense oligonucleotides or gene therapy for *DMD* gene replacement using viral vectors are being studied to emulate a less severe, BMD-like phenotype [6]. However, regulatory agencies are unlikely to approve any treatment without an objective methodology to evaluate patient outcomes. Therefore, the precise quantification of dystrophin in muscle tissue is becoming an essential biomarker to monitor the efficiency of these therapies [7], especially since the measurement of dystrophin in individuals with DMD is extremely challenging due to its near-complete absence [8].

To date, several methods for measuring dystrophin expression have been reported, including immunohistochemistry analysis (IHQ), immunofluorescence (IF), western blot (WB) [8], mass spectrophotometry [9], and enzyme-linked immunosorbent assay [10], but none allow highly accurate quantification of dystrophin. Of the above-mentioned techniques, the most commonly used for dystrophin measurement are IF and WB [7,11,12]. Several protocols for these techniques have been tested and studied to improve dystrophin quantification [13]. IF allows assessment of different areas of the tissue, facilitating the removal of adipose tissue, necrotic fibres and fibrosis, focusing exclusively on the measurement of dystrophin from areas of interest. Additionally, in IF, unlike in WB, dystrophin normalization with housekeeping protein (HKP) is not mandatory to interpret results, as the amount of dystrophin studied in the tissue is not manipulated and, consequently, a loading control is not necessary. This makes it possible to determine which fibres produce protein and quantify it, but this is often done based on subjective evaluation of fluorescence intensity. WB is useful for assessing protein molecular weight, allowing easy detection of dystrophin deletions or duplications. However, WB presents some challenges, such as the large amount of sample required, protein degradation, and technical difficulties due to the large size of dystrophin, as well as the variability between preanalytical tissue lysis conditions and the pitfalls related to the normalization of dystrophin [8]. To accurately measure dystrophin by WB, a HKP is needed, and its selection can be problematic since different HKPs have different ranges of expression that are not always directly proportional to sample load, which can lead to quantification disruptions [14,15]. Moreover, in the absence of a reliable and standardised WB normalization methodology, results can differ between different laboratories; in fact, even within the same laboratory, there may be differences, complicating both quantification and interpretation [8]. Furthermore, when studied by WB, dystrophin usually shows double bands for the full-length protein, which can make quantification even more difficult [13]. In summary, with IF and WB, as either qualitative or semi-quantitative methods, it is difficult to maintain constant criteria for precise protein determination even for expert neuropathologists, which means variation in the assessment of dystrophin. Thus, novel quantification methods are needed to improve the diagnosis and to accurately assess response to treatment. Ideally, new methods should be able to detect subtle dystrophin increases that result from restoring dystrophin expression through read-through therapies, exon skipping therapies, vector-mediated gene therapies or cell therapies [16,17,18]. In this context, standardised results are fundamental since the regulatory agencies would welcome an objective methodology to assess dystrophin upregulation by therapies. In addition, some mutations such as deep intronic variants cannot be detected by routine genetic tests due to the huge genomic size of DMD gene and in these cases, muscle biopsies are needed for diagnosis by identifying a decrease or absence of dystrophin [19,20,21]. There is also a need to improve the quantification for BMD patients with minimal loss of dystrophin, as well as in female carriers, in whom the diagnosis can be very challenging due to minimal muscle histological variability and mosaic expression of dystrophin [22].

In recent years, several dystrophin quantification methods based on image analysis and fluorescence intensity assessment have been described, but these techniques have limitations, mainly the bias introduced by the user in quantification and their time-consuming nature [23]. Therefore, there is a need to implement an automated, reliable and reproducible digital image analysis system for use in diagnosis, treatment monitoring and clinical trials. Confocal microscopy approaches are currently employed for fluorescence quantification, but spectral confocal microscopy has never been used to measure dystrophin. Spectral confocal microscopy systems identify and capture the entire fluorescent emission spectrum of each fluorophore (including the low emissions at both ends and the most intense emission values from the middle of the spectrum), selecting exactly the maximum excitation range in order to increase the sensitivity of fluorescence detection and its subsequent quantification [24,25,26]. In addition, using spectral confocal microscopy, the bleed-through, or crossing and detection of fluorescent emission from a neighbouring channel in the channel of interest, can be removed [12]. Finally, due to the above-mentioned characteristics, and in contrast with alternative techniques such as conventional IF and WB, this approach could be used to better quantify almost-imperceptible fluorescence emissions.

We sought to develop a novel, reproducible and reliable quantification method, using standardised protocols of double immunofluorescence of muscle followed by spectral confocal microscopy image capturing, software processing, and subsequent analysis and quantification. Additionally, to avoid any user bias and to speed up the application of the proposed methodology [23], we automated the image processing using computer vision tools.

## 2. Results

### 2.1. Dystrophin Quantification Methodology Based on Spectral Confocal Microscopy Is Reproducible

To test the reproducibility of the manual method, four independent experiments were performed, each one with the same biopsies from DMD, BMD and healthy control. For each one of the experiments, image analysis was performed as detailed above. Method reproducibility was assessed by intraclass coefficient (ICC), obtaining an ICC = 0.94 with a 95% confidence interval of (0.73, 1). We performed an ICC with three replications as well as Student’s *t*-test to ensure the reproducibility of the methodology.

### 2.2. Spectral Confocal Microscopy Is Highly Sensitive and Can Quantify Very Low Ranges of Fluorescence Intensity

To assess the capacity of spectral confocal microscopy to quantify dystrophin and to prove that this technology is more sensitive than conventional immunofluorescence microscopy, we observed dystrophin IF from muscle samples of patients with DMD, BMD and healthy controls under both types of microscopes. We observed that in severe DMD patients in whom fluorescence signal was almost imperceptible under conventional fluorescent microscopy, values of even 15 a.u. were detected by confocal microscopy, which represents a dystrophin quantity of 0.5% compared to controls (Figure 1).

### 2.3. β-Spectrin Intensity Is Higher in Muscle Biopsies from DMD and BMD Patients Than in Healthy Controls

β-spectrin is a basal sarcolemma protein whose intensity is commonly assumed not to be reduced in muscle dystrophies. Therefore, it has been routinely used as a muscle-membrane integrity control, not only in DMD and BMD, but also in other muscular dystrophies. Thus, for the selection of ROIs in each field, the assessment of sarcolemma integrity, and the detection of the presence of vessels, a basal sarcolemma protein is needed. β-spectrin was labelled and used for this purpose, but it was also observed that its mean intensity was increased in patients compared to controls (DMD show β-spectrin mean intensity of 2346.13 ± 65.88 a.u., BMD of 2379.38 ± 62.43 a.u. and healthy controls of 1832.17± 132.30 a.u.) (Figure 2). There were significant differences in β-spectrin intensity between DMD and controls, as well as between BMD and controls (*p*-value < 0.05). β-spectrin intensity showed no significant differences between muscle samples of DMD and BMD (*p*-value > 0.05).

### 2.4. Fluorescence of Antibodies Detected by Spectral Confocal Microscopy Were Suitable for Separating Patients According to Their Diagnosis

We compared muscle sections of the three different groups (DMD, BMD and healthy controls), and studied ROI mean fluorescence intensity for each of the three antibodies detailed above (NCL-Dys1, NCL-Dys2 and NCL-Dys3). Dystrophin mean intensities using spectral confocal microscopy are shown in Table 1. All antibodies allowed correlation of biopsy fluorescence intensity with the genetic diagnosis of the patient (Figure 3). Although the fluorescence results showed that the three tested antibodies could be used to separate patients according to diagnosis (*p*-value < 0.05), there were differences between their individual behaviours. NCL-Dys1 and NCL-Dys2 showed higher mean fluorescence dispersion in DMD, BMD and healthy controls, being NCL-Dys1 antibody for which the fluorescence results show most dispersion and NCL-Dys3 the one showing least dispersion. By diagnostic group, DMD patients had the least dispersion and healthy controls had the most.

### 2.5. Computer Software-Based Results Did Not Significantly Differ from Manual Results

Since manual selection and analysis of ROIs is a slow process and prone to errors and variation in final results, computer-aided diagnosis systems based on machine vision tools offer a valuable alternative. These systems can speed up the process considerably and, importantly, they provide standardized results in all the studied cases. To determine whether automatic ROI capture was reliable and comparable to manual ROI selection, a comparison between the two methods was performed. The results showed no significant differences for any of the three antibodies tested (NCL-Dys1 *p*-value > 0.05, NCL-Dys2 *p*-value > 0.05, and NCL-Dys3 *p*-value > 0.05) (Figure 4 and Figure 5).

## 3. Discussion

In recent years there has been a great effort by laboratories to find dystrophin quantification methods that are accurate and reproducible [27]. The growing number of clinical trials in neuromuscular diseases and particularly in DMD and BMD, as well as the need for assessment of response to treatment, require new methodologies with highly-sensitive quantification of dystrophin [28]. The expression and quantification of dystrophin as a biomarker in DMD and BMD has been explored for years and, although it correlates with disease progression, it remains a challenge for most pathology laboratories [13]. Although WB and IF are currently the main approaches to semi-quantitative methods in muscle biopsies, IF is considered to introduce the least variation, so most innovative approaches and newly-described quantifying methods are based on it [28]. In addition, there are also alternative, developing methods for dystrophin evaluation such as mass spectrometry [9,13] or microRNA studies [29], but these are systemic serum biomarkers [30] and do not identify the protein, its location or its expression. Different approaches for precise quantification have been conducted using IF as their basis: from the creation of a digital mask as a first step to locate fibres and their sarcolemma, allowing the quantification of a high number of entire fibres in a single section [8,31], to the selection of sarcolemma in small ROIs, which allows the creation of small, adapted regions in the sarcolemma whose intensity is studied. Both these approaches could be operator-independent techniques, minimizing operator bias [10]. In addition to the specific quantification methodology, several imaging approaches have been reported in the literature, using confocal microscopy for dystrophin quantification. However, a spectral approach for fluorescent intensity assessment has not been described before.

We proposed the use of spectral confocal microscopy to develop a methodology to precisely quantify dystrophin. With spectral confocal microscopy, we were able to obtain an independent and broad spectrum of fluorescence emission for each antibody in each ROI, allowing evaluation of the entire fluorescent spectrum and identification of the maximum dystrophin fluorescence emission point, ensuring the subsequent quantification used the highest fluorescence for each case. Unlike other fluorescence quantification methodologies that only provide individual fluorescence measurements, the spectral methodology provides a collection of fluorescence measurements as part of a continuous variable and is therefore more robust. This methodology, combined with white laser, optimized the acquisition of fluorochromes, minimizing their photobleaching in tissues, a key factor in quantification studies [32]. Conventional fluorescence microscopes use mercury lamps and narrow band filters that select a wide excitation band; in contrast, commercial confocal systems use a combination of lasers to excite fluorescence at a small number of discrete wavelengths in the visible spectrum. Flexibility in wavelength selection is essential for optimal excitation and detection of fluorescence, and white laser offers the possibility to choose the optimum excitation wavelength [33]. Our proposed spectral confocal microscopy-based method for dystrophin quantification is equally useful for detecting endogenous fluorescence in tissues, and allows readjustment of experimental conditions if needed to avoid autofluorescence. Autofluorescence is a common problem in tissue immunofluorescence techniques, and can mask the specific fluorescence of the antibody, leading to erroneous results [17,34]. Our proposed methodology makes it possible to study the fluorescence behaviour of different antibodies, and detect whether the fluorescent emission of a particular channel is partially blended with another, a phenomenon known as bleed-through, which is especially relevant when performing double or multiple IF. On balance, spectral confocal microscopy is a robust methodology to monitor common side effects of tissue IF, such as autofluorescence or bleed-through, which are important as they could lead to quantification errors. In this regard, the main purpose of using a single sarcolemma ROI instead of creating a full tissue mask, notwithstanding that it was more time consuming, was that the ROI allowed precise positioning in the sarcolemma, resulting in less influence of additional membrane background, such as fibrous or adipose tissue, and consequently more accurate quantification.

In our experiments, we covered three different areas from the whole protein to ensure adequate coverage using three different antibodies: NCL-Dys1 for ROD domain detection, NCL-Dys2 for C-terminus detection, and NCL-Dys3 for N-terminus detection. The choice of these antibodies was based on the fact they had been extensively tested both in research and in DMD and BMD diagnosis, and also because the commercial company provides a diagnosis certificate for them, thus ensuring their robustness and suitability [12]. This, together with the use of spectral confocal microscopy, ensured that the fluorescence studied in each case was unequivocally from the dystrophin in the muscle biopsy. Although our results demonstrated that each antibody individually gave clear results, correctly classifying each patient according to diagnosis, we suggest it might be more appropriate to include the results of all three antibodies to obtain more accurate data for assessment of fluorescence intensity, as they cover different regions of the dystrophin protein.

It was also observed that β-spectrin was overexpressed in biopsies from patients compared to control biopsies, as previously described [18]. Although this protein is essential for ROI positioning, and β-spectrin had occasionally been used for dystrophin normalization [11], according to our results this could lead to underestimation of dystrophin levels, and the subsequent normalized ratios could highlight differences in dystrophin expression between groups. Our results in β-spectrin quantification suggest that this sarcolemma protein is not suitable for normalizing dystrophin intensity levels and, consequently, it might be pertinent to identify a different membrane protein whose expression is not affected in patients with DMD and BMD.

By comparing muscle biopsies of DMD and BMD patients and controls, we demonstrated that spectral confocal microscopy is able to detect minimal traces of residual dystrophin that cannot be detected under conventional IF imaging techniques. In some DMD cases, we even detected dystrophin at 0.5% intensity compared to healthy controls. The use of hybrid detectors with higher sensitivity and a very high dynamic range allowed us to measure extremely low fluorescence signals. The high sensibility of this new technique allows the quantification of protein traces that were absolutely invisible on conventional microscopy, and which could be critical for detecting subtle increases in dystrophin after treatment. Thus, the use of machine-based approaches could be used in the prediction of progression of some diseases. We are developing and applying artificial intelligence tools to improve diagnostic accuracy and identify novel biomarkers to assess response to gene therapy aimed at restoring dystrophin.

We have also described here a new software with multi-channel features to quantify dystrophin automatically. The methodology is based on the manual method, using multidimensional series from spectral confocal microscopy. This software will be highly useful to reduce errors in ROI positioning introduced by the user when done manually, and will also make the whole process less time consuming. It allows extraction of biologically relevant data from images in a reliable, unbiased and rapid way. In the future, we hope to introduce additional functionalities into this software so that our machine learning method allows us to obtain information about fibre characteristics, fibrosis or amount of adipose tissue using the β-spectrin channel or other antibodies—information that is relevant and still performed manually, representing a substantial bottleneck. In conclusion, we have developed an automated, highly-sensitive method for the quantification of dystrophin, which could be applied not only to measure this protein, but also to measure other sarcolemma proteins of the *DAPC*. Thus, the overall results could be integrated and evaluated together for a wider understanding of the interaction of the different *DAPC* proteins. This integration could help to better explain the alterations occurring in other proteins in patients with DMD and BMD and the heterogeneity of their clinical progression.

## 4. Material and Methods

### 4.1. Ethics Statement

This study was carried out in accordance with the recommendations of Sant Joan de Déu Hospital Ethics Committee. Written informed consent was obtained from patients and controls and/or their parents or guardians in accordance with the Declaration of Helsinki. All the samples were stored under strict quality and traceability conditions in Hospital Sant Joan de Déu Biobank. Patients and their parents/guardians were fully informed about the main ethical and legal implications of participating in a research project, and finally informed consent was obtained. All samples including control muscle samples were under the legal umbrella of Sant Joan de Déu Biobank.

### 4.2. Human Skeletal Muscle Samples and Immunofluorescence Analysis

Open muscle biopsies were performed for diagnostic purposes in patients with suspected muscular dystrophy. We selected patients with genetically confirmed DMD diagnosis (*n* = 10) or BMD diagnosis (*n* = 3) and healthy controls (*n* = 6). Healthy control muscles were obtained during orthopedic surgeries. Patient and healthy control details are shown in Appendix A. All patients were aged between 4 and 6 years old, and biopsy tissue surplus was stored in Sant Joan de Déu Biobank.

Muscle biopsies were processed according to standard procedures [27]. Biopsies with transversely-oriented fibres were frozen in isopentane cooled in liquid nitrogen for 2 min and stored at −80 °C until needed. Ten-micron-thick muscle sections were cut, and after 20 min at room temperature (RT) they were then fixed with paraformaldehyde 4% for 7 min at RT. Then were abundantly rinsed with PBS-Tween 0.5% and then blocked for 90 min with PBS-Tween 1%-BSA 8% at RT. Double immunolabelling was performed and the sections were incubated at 4 °C overnight with dystrophin and β-spectrin. Since dystrophin is a large protein, and to ensure proper coverage of the whole protein, we used IgG mouse primary antibodies against three different regions of dystrophin protein: NCL-Dys1, against the rod-like domain; NCL-Dys2, against the carboxyl terminal; and NCL-Dys3, against the amino terminal (Leica Biosystems, Newcastle, England). To ensure the integrity of the muscle membrane and recognise muscular fibres we used an IgG rabbit primary antibody to label β-spectrin, a sarcolemma protein (PA1-46007. Thermofisher, Waltham, MA, USA). After primary antibody incubation, sections were abundantly rinsed with PBS-Tween-Triton and then incubated for 3 h at RT in the dark with 1:500 dilution secondary antibodies: Alexa Fluor 488 IgG anti-mouse and Alexa Fluor Cy5 IgG anti-rabbit (Thermofisher, Waltham, MA, USA). Finally, the sections were abundantly rinsed with PBS-Tween-Triton and mounted using antifade mounting agent ProLong (Molecular Probes, Eugene, OR, USA).

### 4.3. Image Acquisition by Conventional Fluorescence Microscopy

Fluorescence from immunolabelled sections was collected by an oil immersion 63× objective lens and detected by a Leica DFC7000 T camera (Leica Microsystems GmbH, Mannheim, Germany). A mercury metal halide lamp EL6000 was used as the source and the band pass emission filter cube (fluorescein) was used to visualize dystrophins. Image acquisition was done using LAS X (Leica Microsystems GmbH, Mannheim, Germany). For dystrophins excitation a 470/40 blue band pass filter was used. As β-spectrin was labelled with Cy5 fluorophore and it was not available in the conventional fluorescence microscope, the membrane integrity was assessed by confocal imaging.

### 4.4. Image Acquisition by Spectral Confocal Microscopy

Images from immunolabelled sections were acquired using a Leica TCS SP8 confocal laser-scanning microscope (Leica Microsystems GmbH, Mannheim, Germany). Spectral confocal microscopy was performed using a detection unit that allows spectral discrimination using 63× (NA 1.4, oil) Plan-Apochromatic objective. Images were acquired using HyD detectors with high spectral sensitivity and a high signal-to-noise ratio, in combination with a white-light laser. Dystrophins were excited with a white laser at 470 nm and the fluorescence emission spectrum was collected from 485 to 625 nm with 15 nm bandwidth (stepsize = 5.21 nm). β-spectrin excitation was carried out with white laser at 570 nm and fluorescence emissions were captured in the range from 590 nm to 780 nm with the same conditions as above. The generated image stacks were of the dimensions xyλ. Gains and offset were the same for each of the antibodies used in each field at each excitation wavelength and not altered throughout the scanning process.

The variation in intensity of a particular spectral component, encoded using 12 bits, was represented on the screen using a pseudocolour look-up-table. The image resolution was set to 512 × 512 pixels with a pixel size of 0.361 μm^2^, resulting in a field of view of 184.52 μm^2^.

The mean fluorescence intensity of the x-y-λ data sets was measured using Leica application Suite X (LAS X) software. The region of interest (ROI) function of the software was used to determine the spectral signature of a selected area from the image and the software displayed the mean intensity of all pixels within the ROI versus the wavelength.

### 4.5. Manual Image Analysis

For each biopsy, three different fields were studied and 26 ROIs were placed in each field, obtaining a total of 78 elliptical ROIs for each biopsy. Due to the fibre size variability between cases and controls and indeed among cases themselves, as well as the high magnification of the fibres with the 63× objective, we decided to set the number of fields and ROIs instead of the number of fibres. ROIs were manually selected with LASX software using β-spectrin as a guide to select each ROI and to assess the integrity of the membrane. ROIs were defined so that they only included the membrane, discarding sarcoplasm, vessels and adipose or connective tissue. The data obtained for all the spectrum values for the 78 ROIs allowed us to select accurately, and with a high precision, the maximum intensity of fluorescence emission for dystrophin in each case as a single value for the maximum emission peak. Fluorescence measurements were expressed in arbitrary units (a.u.).

### 4.6. Automatic Image Analysis

Since the manual ROI selection and analysis was a slow process with potential for errors, automatic-aided diagnosis systems relying on computer vision tools were considered to offer a valuable alternative. Such systems can significantly accelerate the process and, more importantly, they provide standardised results. We therefore developed computer vision tools to accurately estimate the quantity of dystrophin in the input images. Based on β-spectrin labelling, the procedure identifies a set of small elliptical ROIs in the sarcolemma where the intensity of the dystrophin is later computed.

The selection of the ROIs involved three steps. First, the muscle fibres were detected, then the sarcolemma was located, and finally, we selected the best positions for the ROIs. This approach ensures the tool is valid for the different stages of the disease, even in advanced stages where the sarcolemma may show irregularities and the tissue shows a higher content of connective and adipose tissue, generating misinformative regions in the images.

To detect muscle fibres, we first improved the input β-spectrin image, selecting the wavelength with the highest fluorescence and removing noise using a median filter. Then, the image was binarized and a sequence of morphological operators was used to identify uniform and approximately circular regions, corresponding to the muscle fibres. In the next step, these regions were dilated to identify the areas separating them, where fibre sarcolemma is located. These identified regions were approximated by little ellipses placed accurately in sarcolemma. In a final step, the elliptical regions were analyzed in further detail, identifying potentially valid ROIs. These ROIs were ranked based on their brightness, their orientation with respect to the sarcolemma, and the total pixels laying inside and outside the wall. The 26 best-ranked regions for each field were selected as the ROIs used to compute the concentration of the desired protein.

The whole procedure was carried out in Matlab (MATLAB 2018a, The MathWorks Inc., Natick, MA, USA), including a graphical user interface to facilitate the visualization of the images of the selected ROIs and their modification, if necessary. The interface also facilitates the comparison of the dystrophin concentration plots for different images. β-spectrin fluorescence intensity was also assessed, but it was not included in the definitive results (data not shown) (Figure 6).

### 4.7. Statistical Analysis

Method reproducibility was assessed by intraclass coefficient (ICC). The Shapiro-Wilk test was used to assess the normality of the distribution of the different datasets (DMD, BMD and healthy control), both in the manual and automatic image analysis methods. Wilcoxon and Kruskal-Wallis tests were performed to compare unpaired raw fluorescence values (DYSn) between the different cohorts. Statistical significance was determined as *p* < 0.05. Plots were generated using RStudio version 1.0.143 (R 3.6 R Foundation for Statistical Computing, Vienna, Austria). For manual and automatic image analysis assessment, a two-sample Student *t*-test assuming equal variances was performed to compare the two methodologies.

## Figures and Tables

**Figure 1 ijms-24-06358-f001:**
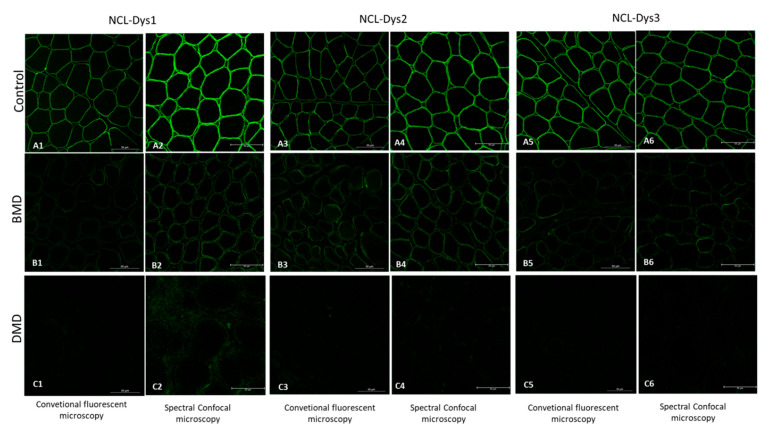
Fluorescent images of the three dystrophin antibodies used (NCL-Dys1, NCL-Dys2 and NCL-Dys3) in control, BMD and DMD muscle biopsies captured with conventional microscopy and spectral confocal microscopy. All the spectral confocal microscopy images (**A2**,**A4**,**A6**,**B2**,**B4**,**B6**,**C2**,**C4**,**C6**) show more intense fluorescence signal than the conventional microscopy images (**A1**,**A3**,**A5**,**B1**,**B3**,**B5**,**C1**,**C3**,**C5**). This is especially relevant in those DMD cases where no fluorescent signal is visible nor quantifiable on conventional microscopy but traces of even 0.5% dystrophin could be detected using spectral confocal microscopy.

**Figure 2 ijms-24-06358-f002:**
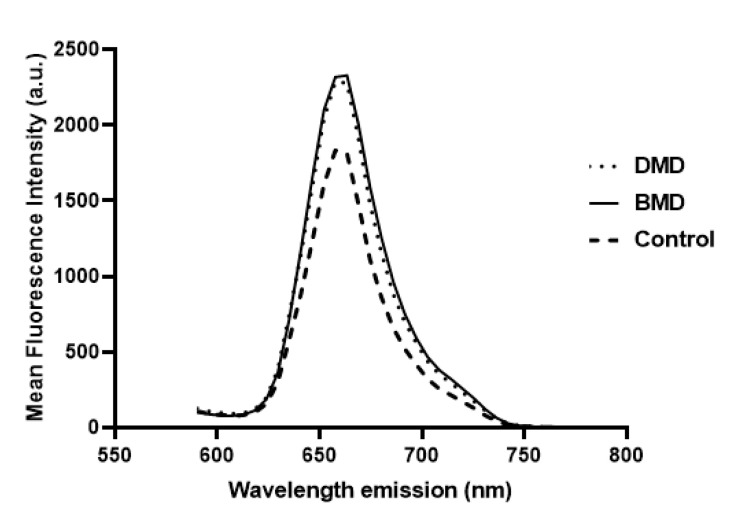
β-spectrin mean fluorescence intensity in relation to emission wavelength computed from β-spectrin spectral stacks of samples. DMD (dotted line) and BMD (solid line) showed higher fluorescence intensity than healthy subjects (dashed line).

**Figure 3 ijms-24-06358-f003:**
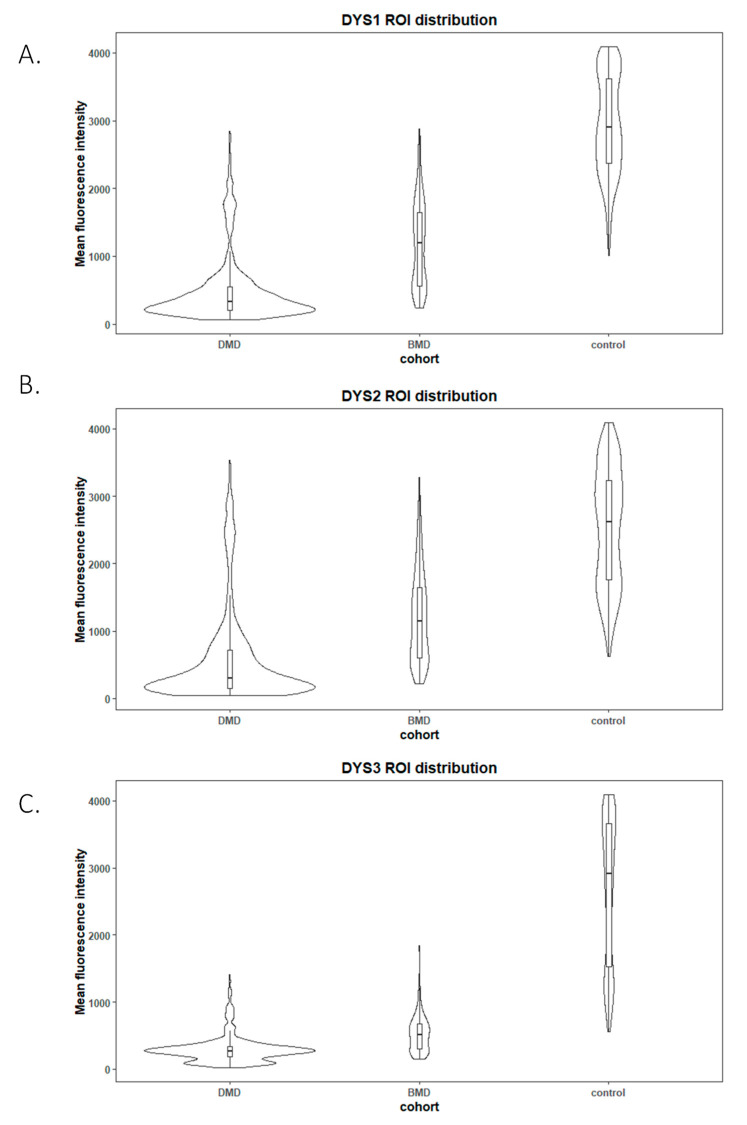
Violin plots of ROI fluorescent intensity distribution from each studied disease group (*n* = 10 DMD, *n* = 3 BMD) and control (*n* = 6) for the different antibodies used NCL-Dys1 (**A**), NCL-Dys2 (**B**) and NCL-Dys3 (**C**). These plots show that tested antibodies were suitable to separate patients according their diagnosis (*p*-value < 0.05). NCL-Dys1 (**A**) was the antibody which showed more fluorescence results dispersion and NCL-Dys3 (**C**) was the one which demonstrated less dispersion. According to diagnosis, DMD was the group which showed less internal ROI dispersion and controls the one with most ROI variability.

**Figure 4 ijms-24-06358-f004:**
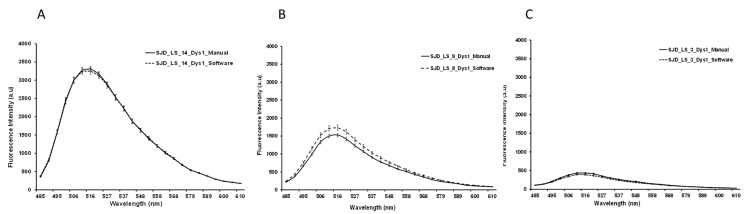
Graphics representing the mean fluorescence intensity spectral of NCL-Dys1 (*y*-axis) and emission wavelength (*x*-axis) at 470 nm excitation wavelength. Representative examples of the comparison between manual (solid line) and automatic quantification (dotted line). Healthy control (**A**). BMD (**B**). DMD (**C**). Results showed no significant differences.

**Figure 5 ijms-24-06358-f005:**
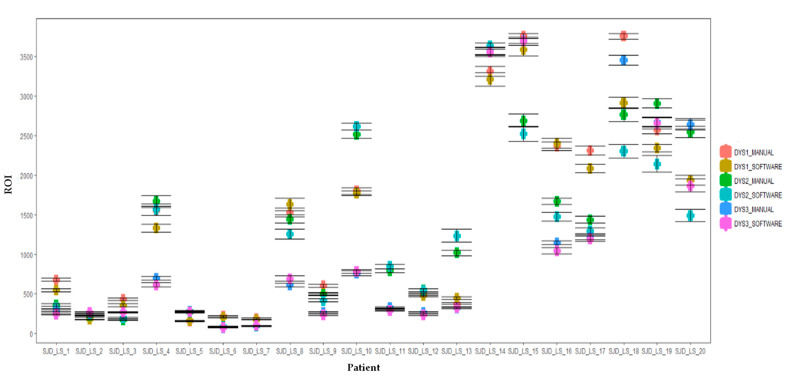
Boxplots of the comparison of manual and automated dystrophin quantification results. Dystrophin levels from each patient with DMD and BMD and healthy control were tested using NCL-Dys1, NCL-Dys2 and NCL-Dys3 and quantified either by a manual ROI positioning approach or with software automated approach. Results showed no significant differences. Errors bars represent standard deviation.

**Figure 6 ijms-24-06358-f006:**
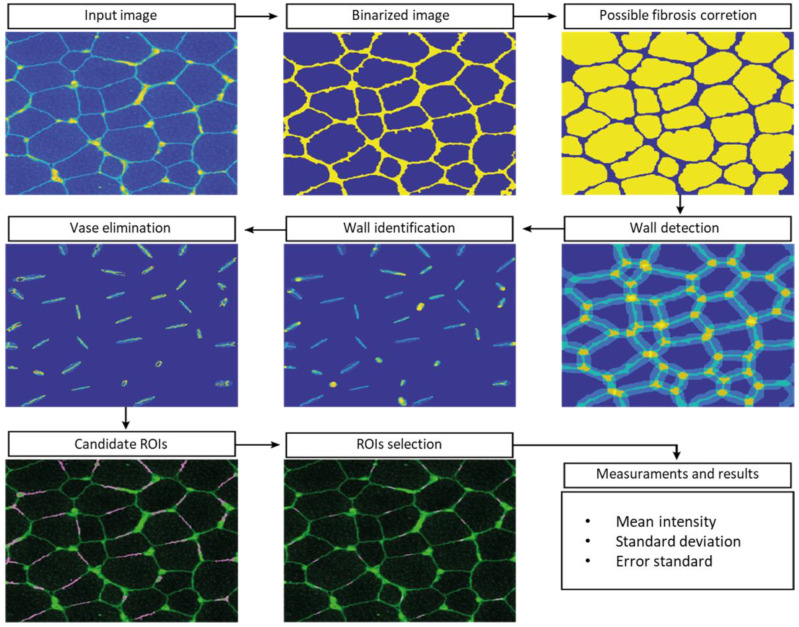
Automatic analysis pipeline on an example image.

**Table 1 ijms-24-06358-t001:** Dystrophin mean fluorescence intensities (a.u.) and standard errors obtained from the 78 ROI of all patients for each pathological group: DMD, BMD and healthy controls. Fluorescent intensities for each antibody separate patients according to their diagnosis.

	Mean Fluorescence Intensity (a.u.)		
	NCL-Dys1	NCL-Dys2	NCL-Dys3
**DMD**	500.8 ± 17.6	590.7 ± 25.3	296.8 ± 7.35
**BMD**	1190.8 ± 42.1	1219.7 ± 44.56	526.02 ± 17.13
**CONTROL**	2982.9 ± 29.3	2732.214 ± 28.4	2920.4 ± 30.91

## Data Availability

The data presented in this study are available on request from the corresponding author. The data are not publicly available due to ethical issues.

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
