# Peer review of "Innovative Computerized Dystrophin Quantification Method Based on Spectral Confocal Microscopy"

_ijms, 2023, doi:10.3390/ijms24076358_

Round 1
Reviewer 1 Report
Wonderful study and manuscript. The findings will be so helpful. Some language edits would be helpful like line 19 - "Several clinical trials..."
Author Response
Thank you for reading the manuscript and for your comments.
We have checked English spelling of the whole manuscript to correct the style.
Reviewer 2 Report
This is a very interesting manuscript that could be useful in tackling the problem of dystrophin quantification in patient samples. Determining the absolute levels of dystrophin is an important and challenging problem and new developments in the field with improved sensitivity are warranted. Please find my comments on this manuscript-
1. Please comment if spectral fluorescence microscopy has been used for quantification of any of the proteins before. If yes, please provide the reference.
2. It is not very clear from the manuscript- how spectral data was used for quantification of mean fluorescence intensity. Does figure 3 data represent MFI at a single wavelength or a cumulative value from fluorescence at different wavelengths. Please elaborate in section 2.4, specifically focusing on how data at different wavelengths was transformed to give a single value for mean fluorescence intensity.
3. Inappropriate use of Beta-spectrin as control in most studies has been discussed in the manuscript. The data from this study is also mentioned in the results section 3.3 but is not represented as a figure. Please provide the data as a separate figure either in the main section or supplementary data.
4. Please provide the legends for Table 1. Provide explanation for the values after the comma. Also correct the mistake where “BMD” is mentioned as “DMB”
5. Please change the Y axis notation in figure 3. ROI is misleading as the y axis should represent the fluorescence intensities. Also mention the subsections of Figures. For example, figure 3 has three panels which should be labelled as 3A, 3B and 3C and mentioned in the figure legend. Please do the same for Figure 4 as well.
6. The comparison for the automated and manual selection should be represented similarly. Please provide the violin plots for automated selection of ROIs similar to figure 3, which should be included in the main text and not supplementary data.
Author Response
- Please comment if spectral fluorescence microscopy has been used for quantification of any of the proteins before. If yes, please provide the reference.
Thank you for your comment, we have revised the bibliography and we don’t have found any reference about the use of spectral microscopy in protein quantification in Duchene Muscular dystrophy, but it has been used in other fields, for example these two references below:
- Autofuorescence of stingray skeletal cartilage: hyperspectral imaging as a tool for histological characterization. Júlia Chaumel, María Marsal, Adrián Gómez‑Sánchez, Michael Blumer, Emilio J. Gualda, Anna de Juan, Pablo Loza‑Alvarez, Mason N. Dean. Discover Materials. 1:16, 2021.
- Evaluation and quantification of spectral information in tissue by confocal microscopy. Journal of Biomedical Optics ·2012.
- It is not very clear from the manuscript- how spectral data was used for quantification of mean fluorescence intensity. Does figure 3 data represent MFI at a single wavelength or a cumulative value from fluorescence at different wavelengths. Please elaborate in section 2.4, specifically focusing on how data at different wavelengths was transformed to give a single value for mean fluorescence intensity.
Yes, we agree with your comments. The mean fluorescence intensity corresponds to the maximum emission of the functions linked to the fluorochrome that we have used to mark the protein for the performance of the statistics. We have made some modifications in the manuscript (line 204): The data obtained for all the spectrum values for the 78 ROIs, allowed us to select accurately and with a high precision the maximum intensity of fluorescence emission for dystrophin in each case as a single value. Fluorescence measurements were expressed in arbitrary units (a.u.).
- Inappropriate use of Beta-spectrin as control in most studies has been discussed in the manuscript. The data from this study is also mentioned in the results section 3.3 but is not represented as a figure. Please provide the data as a separate figure either in the main section or supplementary data.
Thank you for this comment. We agree with you and we have made new figure (figure 3) where Beta-spectrin data is shown. We have made a comparative graphic with Beta-spectrin mean fluorescence intensity in relation with emission wavelength (average curves) computed from spectrin lambdascan stacks of samples. DMD (dotted line) and BMD (solid line) showed higher fluorescence intensity than healthy subjects (dashed line).
We have unified the values of text data for better comprehension. In the original text, we had used the data referring to the increase in B-spectrin with respect to the patients compared to the healthy controls. By incorporating the new figure, we have changed the increase data for the maximum emission value to unify the criteria between the text and the graph.
- Please provide the legends for Table 1. Provide explanation for the values after the comma. Also, correct the mistake where “BMD” is mentioned as “DMB”.
Yes, we agree with your comments. We stated in table 1 the dystrophin mean fluorescence intensities and standard errors obtained from the 78 ROI of all patient for each pathological group: DMD, BMD and healthy controls. Fluorescent intensities for each antibody separate patients according to their diagnosis. We have had some typographical error that we have corrected.
- Please change the Y axis notation in figure 3. ROI is misleading as the y axis should represent the fluorescence intensities. Also mention the subsections of Figures. For example, figure 3 has three panels which should be labelled as 3A, 3B and 3C and mentioned in the figure legend. Please do the same for Figure 4 as well.
We have added a new figure (spectrin levels) and figure 3 has become figure 4. In this case, we have made the changes suggested by the reviewer as we believe the figure is better understandable. We added the legend of the figure 4 and 5.
Figure 4. Violin plots of ROI fluorescent intensity distribution from each studied disease group (n= 10 DMD, 3 BMD) and control (n=6) for the different antibodies used A. NCL-Dys1, B.NCL-Dys2 and C. NCL-Dys3. These plots show that tested antibodies were suitable to separate patients according their diagnosis (p-value < 0.05). NCL-Dys1 (A) was the antibody which show more fluorescence results dispersion and NCL-Dys3 (C) was the one which demonstrates less dispersion. According to diagnosis, DMD was the group which showed less internal ROI dispersion and controls the one with most ROI variability.
- The comparison for the automated and manual selection should be represented similarly. Please provide the violin plots for automated selection of ROIs similar to figure 3, which should be included in the main text and not supplementary data.
Thanks for the comment, but we believe that the curve graph is better understood than the violin plots in this specific case. The graphs of figure 4 and 5 show different approaches, in the case of figure 4, it is showing us the distribution of each ROI according to the diagnosis (DMD, BMD and healthy controls), while in the case of figure 5 it is comparing two calculation methodologies, manual quantification and automatic. In this sense, we have improved Figure 5. In addition, we move the supplementary figure 1 to figure 6.

Reviewer 3 Report
Review of ijms 2074659
Dystrophin quantification by confocal spectral microscopy
This paper describes a computerised method for dystrophin quantification using confocal spectral microscopy in muscle biopsies from DMD & BMD patients, compared with controls. The authors propose that this method and its computerised automation may have advantages for efficiency and standardisation in quantifying very low levels of dystrophin (anticipated in future in DMD treatment trials), and in differentiating mild reductions from normal in the diagnosis of mild cases of BMD (presumably ones who have no immediately detectable dystrophin gene variant, or have one of uncertain significance).
The paper seems very well-written, although for future purposes of comparison requires more detail about the subjects and controls, which can mostly be achieved with an additional Table.
Points needing attention are :
1. Lines 96-7,100: Introduction:
The authors suggest that improved dystrophin quantification will benefit diagnosis, particularly as some deep intronic mutations can be hard to detect by routine techniques, or diagnosis can be difficult in some female manifesting carriers, quoting references from 2013, 2015, 2018. With newer sequencing techniques now more standard, is this still so, or has the role of diagnostic biopsy become more to assist in the interpretation of variants of unknown significance ?
2. Lines 136-139: Section 2.2:
i) Were the DMD & BMD patient biopsies needle or open biopsies ?. Please indicate. Also please state whether the controls were the same (needle or open biopsy).
ii) The authors must indicate what type of patients were the healthy controls, and how they came to get muscle biopsies – ie. was this during surgery for some other condition, and if so what ? Also, please confirm that the controls were all male (eg. '...healthy male controls (n=6*)...'.
iii) There is a discrepancy between the number of controls mentioned here (n-6) and the number referred to in the Legend to Figure 3 (n=7) (line 305). Please make these consistent.
3. Line 144: Typo:
‘They were then ….’ rather than ‘Then were….’
4. Lines 276-281 Section 3.3
Please give the p value also for the combined comparison of DMD + BMD (621) vs Controls (476), assuming that the p<0.05 refers to each of DMD vs controls and BMD vs controls, as separate comparisons. Please also indicate here that the same observation has been seen by others. Eg. ‘…compared to controls (621 ± 39 a.u. versus 476 ± 33 278 a.u.) (p-value = 0.0…), as previously similarly noted by others (18).
5. Line 284 Heading to section 3.4
Since the previous paragraph refers to beta-spectrin; the heading here needs to say '…mean dystrophin fluorescence intensity values...'
6. Line 288-9
In addition to their summary Table (Table 1), the authors need to include a table similar to the Table1 of ref18 (Sardone et al) which lists the individual sample results, giving the subject's diagnostic category (and for controls, the reason for the biopsy), age, sex, the muscle biopsied, the dystrophin mutation, and also the mean (+/- spread) value for dystrophin intensity with each antibody in that subject. This additional Table could be included as supplementary material if preferred.
7. Table 1 Line 297 Typo
The 2nd row of Table 1 should be ‘BMD’, not ‘DBM’.
8. Line 305, Legend to Figure 3.
Control number: n=7 (or n=6 ?). Needs to be consistent with line 138.
9. Line 306-7.
Better to write: ‘…antibodies were suitable for separating patients according to their diagnosis’
10. Line 307-9.
Better to write: ‘…for which the fluorescence results show most dispersion, and NCL-Dys3 was the one showing least dispersion. Based on diagnosis, DMD was the group which showed least internal ROI dispersion, and controls the one with most ROI 309 variability.
11. Line 319 Section 3.5
Supplementary Figure 1 needs a Legend.
12. Lines 321-6 Figure 4 and its Legend
i) It is confusing as to in what way these are representative examples. Is it that only Dys1 is shown, and Dys-2 & 3 are similar; or is it that each of these are plotted from only a few ROI; or is each one from one patient ? Please clarify in the Legend.
ii)Also: the 4 graphs need labelling (? A,B,C ?) - but then why are there 4 plots rather than 3?
iii) Lines 324-5
The authors need to define here that it is the complete set of results that shows no significant differences between manual and automatic for any of the 3 dyes tested, rather than just the represented illustrative examples from Figure 4.
Author Response
- Lines 96-7,100: Introduction:
The authors suggest that improved dystrophin quantification will benefit diagnosis, particularly as some deep intronic mutations can be hard to detect by routine techniques, or diagnosis can be difficult in some female manifesting carriers, quoting references from 2013, 2015, 2018. With newer sequencing techniques now more standard, is this still so, or has the role of diagnostic biopsy become more to assist in the interpretation of variants of unknown significance?
Thank you for this comment. We agree with you and now we added a new reference (21. Lu X, Han C, Mai J, Jiang X, Liao J, Hou Y, et al. Novel Intronic Mutations Introduce Pseudoexons in DMD That Cause Muscular Dystrophy in Patients. Front Genet. 2021;12(April):1–8.). Due to the big genomic size of the DMD gene, even for new molecular techniques, it can be difficult to cover correctly the gene, for this reason, histopathological and immunofluorescence techniques on muscle tissue are helpful in these cases.
In addition, muscle biopsy is also useful to assess mutations of uncertain pathological significance. For this reason, we believe it is important to have methods that allow adequate quantification of the different membrane proteins involved in muscular dystrophies.
- Lines 136-139: Section 2.2:
- i) Were the DMD & BMD patient biopsies needle or open biopsies ?. Please indicate. Also, please state whether the controls were the same (needle or open biopsy).
- ii)The authors must indicate what type of patients were the healthy controls, and how they came to get muscle biopsies – ie. was this during surgery for some other condition, and if so what ? Also, please confirm that the controls were all male (eg. '...healthy male controls (n=6*)...'.
iii) There is a discrepancy between the number of controls mentioned here (n-6) and the number referred to in the Legend to Figure 3 (n=7) (line 305). Please make these consistent.
Thank for your comment. We added and corrected the requested information.
- Open muscle biopsies were performed for diagnostic purposes in patients with suspected muscular dystrophy.
- Healthy controls muscles were obtained during orthopedic surgeries.
- The number of control was 6. We changed it in the new figure 4.
- Line 144: Typo:
‘They were then ….’ rather than ‘Then were….’
We agree with you and we have modified the sentence.
- Lines 276-281 Section 3.3
Please give the p value also for the combined comparison of DMD + BMD (621) vs Controls (476), assuming that the p<0.05 refers to each of DMD vs controls and BMD vs controls, as separate comparisons.
Yes, we agree with your comments, and we have made new figure (figure 3). We have unified the values of text data for better comprehension. In the original text we had used the data referring to the increase in B-spectrin with respect to the patients compared to the healthy controls. By incorporating the new figure, we have changed the increase data for the maximum emission value to unify the criteria between the text and the graph. We added the stadistical data as p-value < 0.05, as previously publicized similarly by others
- Line 284 Heading to section 3.4
Since the previous paragraph refers to beta-spectrin; the heading here needs to say '…mean dystrophin fluorescence intensity values...'
Thank you for this comment. We have changed the heading. Fluorescence of antibodies detected by spectral confocal microscopy were suitable for separating patients according to their diagnosis.
- Line 288-9
In addition to their summary Table (Table 1), the authors need to include a table similar to the Table1 of ref18 (Sardone et al) which lists the individual sample results, giving the subject's diagnostic category (and for controls, the reason for the biopsy), age, sex, the muscle biopsied, the dystrophin mutation, and also the mean (+/- spread) value for dystrophin intensity with each antibody in that subject. This additional Table could be included as supplementary material if preferred.
Thanks for this interesting comment. We include a new supplementary table listed as 1 with all the information requested.
- Table 1 Line 297 Typo
The 2nd row of Table 1 should be ‘BMD’, not ‘DBM’.
We have had some typographical error that we have corrected.
- Line 305, Legend to Figure 3.
Control number: n=7 (or n=6 ?). Needs to be consistent with line 138.
The number of healthy control used (n) was 6. We changed it in the new figure 4.
- Line 306-7.
Better to write: ‘…antibodies were suitable for separating patients according to their diagnosis
Thank you for this comment. We have changed the heading. Fluorescence of antibodies detected by spectral confocal microscopy were suitable for separating patients according to their diagnosis.
- Line 307-9.
Better to write: ‘…for which the fluorescence results show most dispersion, and NCL-Dys3 was the one showing least dispersion. Based on diagnosis, DMD was the group which showed least internal ROI dispersion, and controls the one with most ROI 309 variability.
Thanks for this comment. We changed the text. NCL-Dys1 and NCL-Dys2 showed higher mean fluorescence dispersion in DMD, BMD and healthy controls, NCL-Dys1 for which the fluorescence results show most dispersion, and NCL-Dys3 was the one showing least dispersion. By diagnostic group, DMD patients had the least dispersion and healthy controls had the most.
- Line 319 Section 3.5
Supplementary Figure 1 needs a Legend.
We moved the supplementary figure 1 to figure 6. The legend is the following. Figure 6: Boxplots of the comparison of manual and automated dystrophin quantification results. Dystrophin levels from each patient with DMD and BMD and healthy control were tested using NCL-Dys1, NCL-Dys2 and NCL-Dys3 and quantified either by a manual ROI positioning approach or with software automated approach. Results showed no significant differences. Errors bars represent standard deviation.
- Lines 321-6 Figure 4 and its Legend
i) It is confusing as to in what way these are representative examples. Is it that only Dys1 is shown, and Dys-2 & 3 are similar; or is it that each of these are plotted from only a few ROI; or is each one from one patient ? Please clarify in the Legend.
ii)Also: the 4 graphs need labelling (? A,B,C ?) - but then why are there 4 plots rather than 3?
Thanks for the interesting comment. We have a new figure listed as 5 more clear with only three graphs instead of four. The new legend is the following:
Figure 5. Graphics representing the mean fluorescence intensity spectral of NCL-Dys1 (y-axis) and emission wavelength (x-axis) at 470 nm excitation wavelength. Representative examples of the comparison between manual (solid line) and automatic quantification (dotted line). Healthy controls (A). BMD (B). DMD (C). Results showed no significant differences.
iii) Lines 324-5 The authors need to define here that it is the complete set of results that shows no significant differences between manual and automatic for any of the 3 dyes tested, rather than just the represented illustrative examples from Figure 4.
We totally agree with you. We have added a new figure listed as 5. We think is better for the compression the results. In the main text the p-valour are listed.
The results showed no significant differences for any of the three fluorochromes tested (NCL-Dys1 p-value > 0.05, NCL-Dys2 p-value > 0.05, and NCL-Dys3 p-value > 0.05
